# Serum Amyloid A Concentrations in Young Japanese Black Cattle: Relationship with Colostrum Intake and Establishing Cut-Off Concentrations

**DOI:** 10.3390/ani15091239

**Published:** 2025-04-28

**Authors:** Urara Shinya, Osamu Yamato, Yuka Iwamura, Tomohiro Kato, Yuhei Hamada, Oky Setyo Widodo, Masayasu Taniguchi, Mitsuhiro Takagi

**Affiliations:** 1Soo Veterinary Clinic, Kagoshima A.M.A.A., Soo 899-8242, Japan; shinya-u@nosai46.jp (U.S.); iwamura-y@nosai46.jp (Y.I.); kato-t@nosai46.jp (T.K.); hamada-y@nosai46.jp (Y.H.); 2Clinical Laboratory Training Center Eastern Laboratory, Kagoshima A.M.A.A., Soo 899-8242, Japan; 3Joint Faculty of Veterinary Medicine, Kagoshima University, Kagoshima 890-0065, Japan; osam@vet.kagoshima-u.ac.jp; 4Division of Animal Husbandry, Faculty of Veterinary Medicine, Universitas Airlangga, Surabaya 60115, Indonesia; oky.widodo@fkh.unair.ac.id; 5Joint Graduate School of Veterinary Sciences, Yamaguchi University, Yamaguchi 753-8515, Japan; masa0810@yamaguchi-u.ac.jp; 6Joint Faculty of Veterinary Medicine, Yamaguchi University, Yamaguchi 753-8515, Japan

**Keywords:** Japanese Black calves, serum amyloid A, mammary-associated amyloid A, colostrum, health care

## Abstract

The results of this study provide serum amyloid A (SAA) cut-off concentrations from 30 days after birth through the growing period in Japanese Black cattle and suggest the potential for a more objective assessment of prognosis and treatment efficacy in diseased cattle from birth to growth, a period with a high incidence of disease.

## 1. Introduction

Serum amyloid A (SAA) is an acute-phase protein (APP) that is biosynthesized in the liver and serves as an indicator of acute inflammation in humans [1,2] and some animals, including cattle [3,4,5,6]. As an APP, SAA levels rise rapidly during inflammation and decline when inflammation subsides. These characteristics have been used to determine treatment strategies and prognoses. There have been reports on the clinical application of SAA in cattle, including dairy cattle [7,8] and Japanese Black (JB) suckling calves [9]. However, no studies have examined the clinical application of SAA in JB breeding cattle. To address this gap, the authors recently reported a cut-off value of 6.5 mg/L for SAA concentrations in JB beef cows [10]. Furthermore, we demonstrated the clinical utility of our established cut-off value in farm practice, showing its applicability in diagnosing various diseases of JB cattle, such as postpartum metabolic disorders, bronchitis, and enteritis [11].

Preventing infectious diseases, particularly diarrhea and respiratory diseases, is a major concern in calf management as calves, especially JB calves since they lack fully differentiated innate and adaptive immune systems during the pre-weaning period [9,12]. The neonatal stage is a critical period for calves to adapt to their environment, with immune system development being a key part of this process [13]. These immune defenses gradually develop during weaning [3]; thus, maintaining calf health during the suckling period is essential to ensure calf survival after weaning.

Despite advancements in vaccination protocols, herd management, and treatment strategies, pre-weaning diseases such as diarrhea and pneumonia continue to cause considerable economic losses. Additionally, during weaning, calves are exposed to several stressors; they are more susceptible to disease and tend to lose weight due to reduced feed intake [14,15,16]. Thus, weaning (at approximately 3 months for JB cattle) is a critical step in calf management and is implemented using various methods at beef and dairy farms [14,17]. Therefore, measuring SAA concentrations during the growth period from birth to weaning at JB production sites could enable the monitoring of inflammatory dynamics in vivo, which is highly beneficial for disease diagnosis, assessing treatment efficacy, and determining prognoses.

Bovine SAA proteins include SAA 1, 2, 3, and 4, with SAA 1 and 2 serving as classical acute-phase response proteins [7,18]. The SAA 3 isoform, known as mammary gland-associated SAA (MAA) of bovine and female cattle, is highly concentrated in the colostrum [7,19] and milk [7,20]. In bovine colostrum, there are highly elevated levels of extrahepatically secreted mammary-associated SAA 3, especially during the first few days after calving [13]. The concentrations of different APPs, including SAAs, in heifers change during the first few weeks of life and are termed “age-related changes in APP” [13]. Colostrum intake has been suggested to influence SAA concentrations, as calves exhibit low SAA concentrations before colostrum consumption, followed by an increase within the first 24 h after birth [13,21,22]; however, no direct transfer of SAA isoforms in calves has been found when measuring colostrum and circulating calf SAA isoforms [13,21]. Instead, it has been suggested that calves synthesize adult SAA isotypes independently, with circulating SAA originating from factors related to the birth process or colostrum rather than disease-related mechanisms [21,23].

Additionally, it has recently been reported that colostrum SAA and circulating calf SAA levels are inversely associated during the first week of life, suggesting that the calf inflammatory response is activated sooner when less protection from colostrum SAA is available [13], which may indicate a protective role for colostrum SAA shortly after birth. Furthermore, regarding the concentration of circulating SAA in clinically healthy dairy calves, it has been concluded that some age-related changes in SAA concentrations occur independently of clinical disease and are most likely caused by physiological processes or subclinical infections [13]. Thus, the relatively high SAA concentrations after birth emphasize the importance of considering the age of the calf when using this protein as a disease marker [21,23].

In postnatal Holstein calves, SAA concentrations fluctuate during the first few weeks of life and stabilize after the third week; they are also influenced by colostrum intake, environmental factors, and age [13,21]. In our previous study on adding heat-killed bacteria to milk replacer in JB calves, the SAA concentrations and daily health status were monitored, starting immediately after colostrum intake and continuing until a month after weaning (at approximately 3 months of age). The results showed that the average SAA concentration in clinically normal JB calves tended to be higher than that in JB cows (6.5 mg/L), and the average SAA concentration in clinically normal JB calves was higher than that in JB cows [9,12]. However, during daily clinical evaluations, applying the cut-off values established for SAA in JB breeding cows proved challenging for assessing calves, even during JB clinical activities spanning the calving and growing seasons. To date, no studies have assessed the SAA concentrations in JB cattle throughout their growing period, spanning birth to 10 months of age. This study aimed to address this gap by establishing cut-off values for JB calves, utilizing a clinical database of SAA concentrations derived from real-world clinical cases in our practice.

## 2. Materials and Methods

### 2.1. Ethical Statement

This study was conducted in accordance with the regulations for the protection of experimental animals and the guidelines of Yamaguchi University, Yamaguchi, Japan (No. 40, 1995; approved on 27 March 2017). Informed consent was obtained from all the farmers involved in this study.

### 2.2. Young Japanese Black (JB) Cattle

This study included JB calves from the suckling period to the growing period (up to 10 months of age) under the jurisdiction of the Soo Veterinary Clinic of the Soo Agricultural Mutual Aid Union, Kagoshima Prefecture, Japan. The healthy young JB cattle included in this study were of various ages and showed no visible signs of clinical disease. They underwent routine metabolic profile tests (health checks) conducted at our clinic, with blood biochemistry analyses performed in the laboratory.

In contrast, diseased animals were examined upon request by farmers for diagnosis. All animals were housed indoors, with the feeding and management practices varying by herd. Suckling calves were fed either their mother’s milk or a milk replacer, while post-weaning growing cattle were primarily fed roughage and a growing concentrate.

Blood samples from healthy animals were collected from the jugular vein of each calf after the morning feeding. For diseased animals, blood samples were collected in the same manner from affected calves, following a veterinarian’s examination and clinical diagnosis. In this study, blood sampling from all diseased calves was conducted between 1 and 4 days after disease onset. After collection, the blood samples were immediately placed on ice and transported to the laboratory. They were centrifuged at 500× *g* for 15 min at room temperature, and the resulting serum samples were frozen at −30 °C until further analysis.

### 2.3. Blood Analysis

Complete blood counts were performed using an automatic blood cell counter (F-820; Sysmex, Kobe, Japan). Serum biochemical analysis was conducted using the Labospect 7180 autoanalyzer (Hitachi, Tokyo, Japan) to measure the following parameters: blood sialic acid, glucose, free fatty acid, total cholesterol, total protein (TP), albumin, bilirubin, urea nitrogen, aspartate aminotransferase, γ-glutamyltransferase (GGT), calcium, magnesium, iron, and inorganic phosphorus levels.

The SAA concentrations were measured using an automated biochemical analyzer (Labospect 7180 autoanalyzer or Pentra C200; HORIBA ABX SAS, Montpellier, France) with a specialized SAA reagent for animal serum or plasma (VET-SAA ‘Eiken’ reagent; Eiken Chemical Co., Ltd., Tokyo, Japan). The SAA concentrations were calculated based on a standard curve generated using a calibrator (VET-SAA calibrator set; Eiken Chemical Co., Ltd., Tokyo, Japan).

The MAA concentrations were determined via an enzyme-linked immunosorbent assay (ELISA) using a COW SAA 3 ELISA kit (Veterinary Biomarkers, Inc., West Chester, PA, USA). Protein electrophoresis was performed using a fully automated capillary electrophoresis system (Sebia, Tokyo, Japan), and the electrolytes were measured with an electrolyte analyzer (Techno Medica Co., Ltd., Kanagawa, Japan). Serum vitamin A and vitamin E levels were measured using a high-performance liquid chromatography system (Shimadzu, Kyoto, Japan) to evaluate changes in vitamin depletion during the experimental period.

#### 2.3.1. Experiment 1

A preliminary trial was conducted to determine the variation in blood SAA and MAA (SAA 3: an isozyme of SAA considered to be derived from milk) concentrations in JB calves before and after colostrum intake. For this purpose, SAA and MAA concentrations were measured in blood samples obtained from two clinically healthy JB calves on two separate occasions: immediately after birth and prior to colostrum intake, and one day following colostrum intake. Additionally, the SAA concentrations of 6 non-colostrum-consuming calves (blood collected within 1 h of birth) were compared to those of 12 colostrum-consuming calves (blood collected within 48 h of birth).

#### 2.3.2. Experiment 2

A total of 331 clinically healthy animals and 164 diseased animals that underwent blood examinations at our facility were enrolled. The animals were classified into five groups according to age: 1–29 days, 30–59 days, 60–89 days, 90–119 days, and 120–300 days. The cut-off values for SAA concentration in the healthy and diseased groups for each age group were determined based on the receiver operating characteristic (ROC) curves. The demographic information of the calves included in this study is shown in Table 1, and the detailed diagnoses of the clinically diseased young cattle are shown in Table 2.

Seventy-six clinically healthy JB calves aged 2 to 28 days were classified into four age groups: 2–7, 8–14, 15–21, and 22–28 days old. The mean (± standard deviation; SD) SAA concentration in each group was calculated to provide a reference value for the SAA concentration in 1-month-old JB calves. Additionally, the correlations between SAA concentrations and blood biochemical parameters were analyzed in 70 JB calves aged 1–7 days.

### 2.4. Data Management and Statistical Analysis

The Mann–Whitney U test was employed to compare SAA concentrations between two groups: colostrum-consuming and non-colostrum-consuming JB calves. The cut-off value for SAA concentration for each age group (1–29 days, 30–59 days, 60–89 days, 90–119 days, and 120–300 days) was established using the receiver operating characteristic (ROC) curve. The model’s prediction quality was assessed using the area under the ROC curve (AUC), which was classified as follows: 0.50–0.60 = failed or random; >0.60–0.70 = poor, weak, or low; >0.70–0.80 = moderate, fair, or acceptable; >0.80–0.90 = good; and >0.90–1.00 = very good or excellent. Only excellent AUCs were considered acceptable in this study. In the 1–29 days age group, the AUC value fell within the >0.60–0.70 range (indicating poor, weak, or low performance); therefore, an alternative method was used to determine the reference for the SAA concentration for calves under one month old. For these calves, the SAA concentrations were calculated for the four age subgroups (2–7, 8–14, 15–21, and 22–28 days) for reference purposes. Finally, the correlations between SAA concentrations and other biomarkers (globulin, TP, blood urea nitrogen, and GGT levels) were analyzed using Spearman’s rank correlation coefficient. All statistical analyses were conducted using EZR (Easy R) version 1.68. The results obtained for each herd are expressed as the mean ± SD, and *p*-values less than 0.05 were considered statistically significant.

## 3. Results

### 3.1. Experiment 1

The mean SAA concentration in calves whose blood was collected twice at 1 day of age was 1.8 mg/L before colostrum intake and 53.3 mg/L afterwards. Calves whose blood was collected only once at 1 day of age after colostrum intake exhibited a mean SAA concentration of 46.7 mg/L. At 1 day of age, the MAA concentrations were below the detection limit (≤0.7 mg/L) before colostrum intake, 2.6 mg/L after colostrum intake, and 4.6 mg/L for calves sampled only after colostrum intake. In the calves whose blood was collected twice, the MAA concentration increased only slightly after colostrum intake, whereas the SAA concentration increased rapidly (Figure 1A). SAA concentrations (32.5 ± 16.4 mg/L) were significantly higher (*p* < 0.05) in the colostrum-consuming group compared to those in the non-colostrum-consuming group (3.0 ± 1.1 mg/L) (Figure 1B).

### 3.2. Experiment 2

The ROC curves for each age group are shown in Figure 2. The AUC for the 1–29 days group was 0.630, the sensitivity was 0.513, and the specificity was 0.702, which was unreliable. The AUC for the 30–59 days group was 0.904, the sensitivity was 0.778, and the specificity was 0.875. The AUC for the 60–89 days group was 0.965, the sensitivity was 0.800, and the specificity was 1.000. The AUC for the 90–119 days group was 0.987, the sensitivity was 0.913, and the specificity was 0.987. The AUC for the 120–300 days group was 0.960, the sensitivity was 0.953, and the specificity was 0.838. The cut-off concentrations obtained from the ROC curves were all ≥0.9, indicating high reliability (30–59 days old: 18.5 mg/L; 60–89 days old: 17.7 mg/L; 90–119 days old: 14.4 mg/L; and 120–300 days old: 8.1 mg/L).

The cut-off values decreased with as the age increased (Figure 3, in the blue area). The means ± SDs of the SAA concentrations were as follows: 39.6 ± 24.8 mg/L at 2–7 days of age (*n* = 51), 34.6 ± 8.5 mg/L at 8–14 days (*n* = 6), 13.8 ± 5.4 mg/L at 15–21 days (*n* = 9), and 9.1 ± 3.4 mg/L at 22–28 days (*n* = 10). The reference values at 1 month of age tended to gradually decrease with age (Figure 3, green box and whisker plot). The correlation analysis showed no correlation between SAA concentration and GGT, TP, or globulin levels, which are believed to reflect colostrum absorption (Figure 4). The blood urea nitrogen level (*r* = 0.468, *p* < 0.01) was the only parameter that showed a significant correlation with SAA concentration, whereas no correlation was found between the other parameters and SAA concentration (Figure 4). The results of this study are summarized in Figure 5. No significant correlations were found between SAA concentration and blood sialic acid, glucose, free fatty acid, total cholesterol, aspartate aminotransferase, calcium, magnesium, iron, and inorganic phosphorus levels, complete blood counts, or serum vitamin A and E levels; therefore, the results for these parameters are not presented.

## 4. Discussion

In our previous study on the addition of dead bacterial products to milk replacers for JB calves, SAA concentrations were measured immediately after colostrum intake at birth and up to 3 months of age. The main objective was to monitor the SAA concentrations of the groups with and without the supplement to determine whether there were differences in the in vivo inflammatory status due to the addition of the supplement. The results showed that in JB calves, immediately after birth and after weaning (at 3 months old in that case), the SAA concentrations were considerably higher (day 1: approximately 40.0 mg/L; day 60: approximately 7.0 mg/L) [9,12]. This motivated us to calculate SAA cut-off values for JB calves in this study.

Regarding blood SAA levels in newborn Holstein calves, one of the major factors affecting neonates in the first week of life is colostrum intake, which significantly fluctuates in the first week of life and stabilizes after the third week, although it is also influenced by environmental factors and maternal age [13,21]. It has been suggested that SAA isotypes (SAA 3/MAA) in bovine colostrum do not migrate from the calf gut into its systemic bloodstream [21]. Therefore, high MAA concentrations in colostrum are beneficial for the calf, providing local protection without a strong systemic inflammatory response [23].

Furthermore, it has been recently reported that colostrum MAA and circulating calf SAA levels are inversely associated during the first week of life. This suggests that under conditions of low colostrum MAA intake and protection, the calf’s own inflammatory response is activated by pro-inflammatory cytokines migrated from the colostrum, such as interleukin (IL)-6, IL-1β, and tumor necrosis factor-α, increasing the calf SAA concentration. This suggests that colostrum SAA may play a protective role in the first weeks of life [13].

Regarding circulating SAA levels in clinically healthy dairy calves, age-related changes in SAA levels occur independently of clinical disease and are probably caused by physiological factors or subclinical infections [13,21]. Following reports on blood SAA concentrations in Holstein calves, this study aimed to determine the dynamics of SAA concentrations in JB calves, which have juvenile immunity from birth to the pre- and post-weaning periods, and to calculate SAA cut-off concentrations in clinically normal and diseased JB calves under farm conditions.

Experiment 1 was conducted as a preliminary study to determine (1) the relationship between colostrum MAA (SAA 3) concentration and the concentrations of calf MAA and SAA in blood samples, and (2) the screening of SAA concentrations starting from immediately after birth before colostrum intake (within 1 h after birth) to 48 h after birth following colostrum intake. Although this study presents the result of only one case, the concentration of serum MAA was below the limit of measurement (≤ 0.7 mg/L) at 1 day of age (before colostrum intake) and increased to 2.6 mg/L at 1 day of age (after colostrum intake) and 4.6 mg/L for another case. However, the increased MAA concentrations were lower compared to the SAA concentrations (before: 1.80 and 1.70 mg/L; after: 53.30 and 46.90 mg/L, respectively; Figure 3). MAA is produced in the mammary tissues of mammals, including humans and cows, and is secreted into the colostrum [24,25]. The present results indicate, for the first time, that in JB calves, only a slight amount of the MAA in colostrum is absorbed, and that the calf itself is likely to biosynthesize the SAA in its serum, triggered by the ingestion of colostrum proinflammatory cytokines, similar to what was previously reported in Holstein calves. Our results suggest that JB calves, after colostrum intake, biosynthesize SAA on their own as a biological defense mechanism. Recently, immunophysiological functions of SAA have been reported [24,25], suggesting that the SAA response in this study may be one of the biological defense strategies of calves, which are immature compared to mature cows in terms of various biological functions.

The results of Experiment 2 allowed for the determination of cut-off concentrations for SAA at 30–59, 60–89, 90–119, and 120–300 days of age for JB cattle. However, cut-off concentrations could not be determined for the 1–29 days age group because the AUC in the ROC curve was low and unreliable. All the cut-off concentrations for 30–300 days of age were higher than those for established JB breeding cows (6.5 mg/L) and gradually decreased with advancing age. Therefore, including the results obtained in this study, we determined the SAA cut-off concentrations for mature JB cattle from 30 days of age onward.

In contrast, in this study, the mean ± SD of the SAA concentration in 1-month-old (from birth to ≤ 30 days of age) healthy calves, for which cut-off concentrations could not be determined, was calculated as a reference concentration for clinically healthy calves. According to a previous study, the circulating SAA concentrations in 143 dairy calves at 1–7, 8–14, and 15–22 days after birth were 146.6 ± 66.5, 142.3 ± 78.8, and 93.0 ± 61.6 mg/L, respectively [13]. Although the mean SAA concentrations of 2 to 7-day-old and 8 to 14-day-old JB calves were high, at ≥30 mg/L, the mean SAA concentrations tended to decrease with age compared to those reported in previous studies. The SAA concentrations in JB calves seemed to be lower than those in dairy calves. Therefore, it was hypothesized that this difference in SAA concentrations immediately after birth may account for the low immunocompetence of JB calves and the high incidence of disease during the suckling period. Our results indicate that for the clinical use of SAA in 1-month-old JB calves, whose cut-off concentrations cannot be calculated, it is necessary to consider them comprehensively by referring to their daily age, colostrum intake status, and the values of other biochemical parameters, particularly the levels of GGT, TP, and globulin, which reflect colostrum absorption.

Additionally, as the early detection of infections in newborns and premature infants is possible by measuring several inflammation indices in human medicine [26,27], it could be possible to evaluate inflammation by referring to several other acute-phase inflammation indices, such as haptoglobin levels, especially in young JB calves soon after birth, and the reliability would also increase. Using the SAA cut-off concentrations determined in this study for diseased calves would be beneficial in clinical practice for an objective evaluation of prognosis and treatment effectiveness, similar to previously reported results for JB breeding cattle. Furthermore, the measurement of SAA concentrations in young calves after colostrum intake suggests the possibility of evaluating acute inflammation and its biological defense function. However, we were unable to identify their defense functions in this study. In the future, we would like to further investigate SAA concentrations and their biological defense functions in young JB calves.

## 5. Conclusions

In conclusion, our findings show that calves that received colostrum had significantly higher SAA concentrations compared to calves that did not. The SAA concentrations at 1–7 days of age did not correlate with GGT, TP, or globulin levels. The cutoff SAA concentrations in JB calves were 18.5 mg/L at 30–59 days of age, 17.7 mg/L at 60–89 days of age, 14.4 mg/L at 90–119 days of age, and 8.1 mg/L at 120–300 days of age. Furthermore, the elevated SAA concentrations in JB calves ≤ 1 month of age suggest that SAA is not absorbed from colostrum, as observed in Holstein calves. Instead, SAA may be biosynthesized as part of the calf’s own biological defense mechanism. Future research is needed to test the application of these SAA cut-off concentrations for clinical diagnosis and to better understanding their role in improving JB productivity.

## Figures and Tables

**Figure 1 animals-15-01239-f001:**
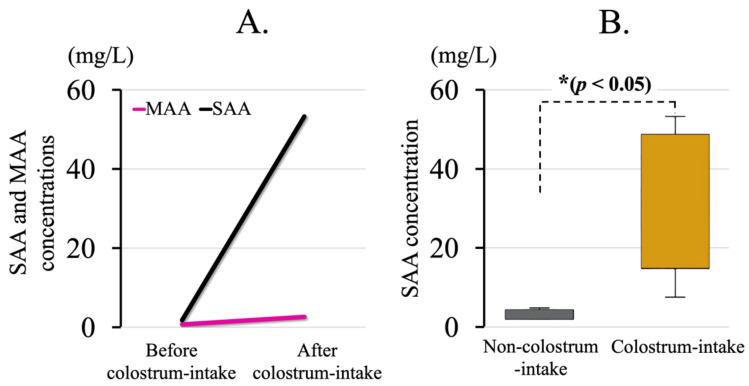
Concentrations of serum amyloid A (SAA) and mammary-associated amyloid A (MAA) in the two different trial groups. (**A**) SAA and MAA concentrations on day 1 before and after colostrum intake in two calves. (**B**) SAA concentrations in colostrum-consuming calves (within 1 h after birth, *n* = 12) and non-colostrum-consuming calves (within 48 h after birth, *n* = 6). *: statistically significant difference (*p* < 0.05).

**Figure 2 animals-15-01239-f002:**
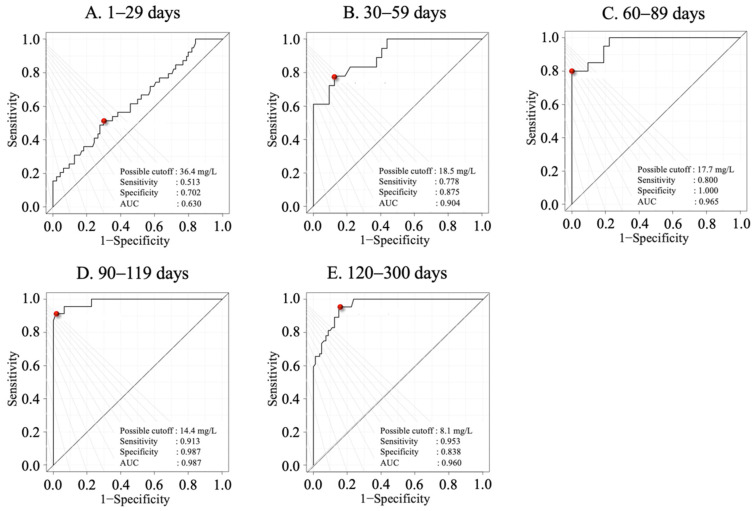
The receiver operating characteristic curves of different young Japanese Black cattle age groups (days) to differentiate healthy and diseased groups according to serum amyloid A (SAA) concentrations: (**A**) 1–29 days of age; (**B**) 30–59 days of age; (**C**) 60–89 days of age, (**D**) 90–119 days of age; and (**E**) 120–300 days of age. Red dot: possible cut-offs.

**Figure 3 animals-15-01239-f003:**
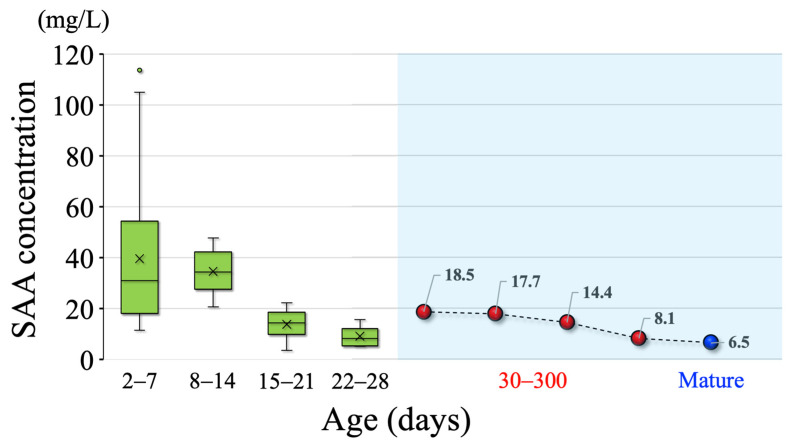
Average serum amyloid A concentrations in clinically healthy 1-month-old Japanese Black (JB) calves (2–7, 8–14, 15–21, and 22–28 days old) and cut-off values for JBs from 30 days of age to matured breeding stage. The young JB cattle in this study were 30–300 days old (red dots). The results for mature breeding cows (blue dot) are from our previously published paper [10].

**Figure 4 animals-15-01239-f004:**
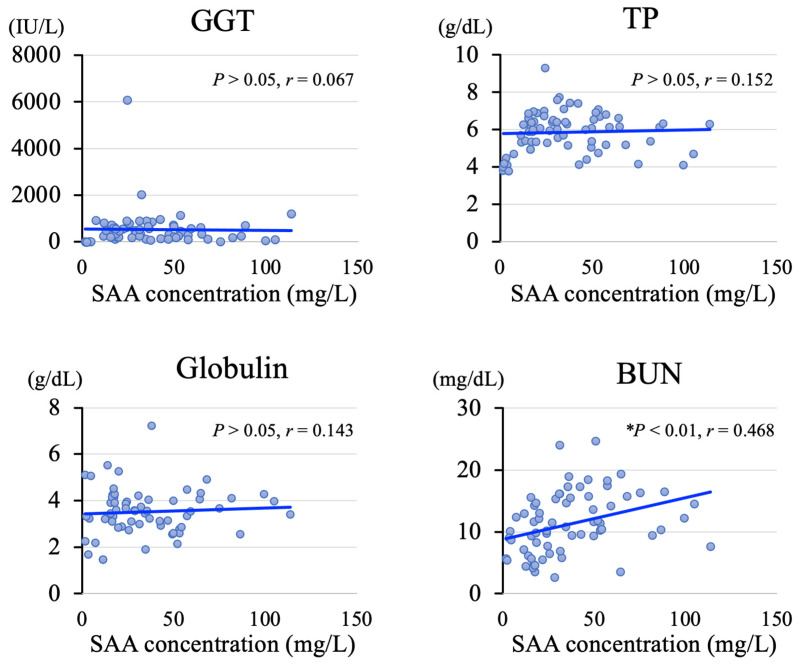
Correlations between SAA concentration and serum biochemical test parameters reflecting colostrum intake in calves after birth. *: statistically significant difference (*p* < 0.05). BUN, blood urea nitrogen; GGT, γ-glutamyltransferase; TP, total protein; SAA, serum amyloid A.

**Figure 5 animals-15-01239-f005:**
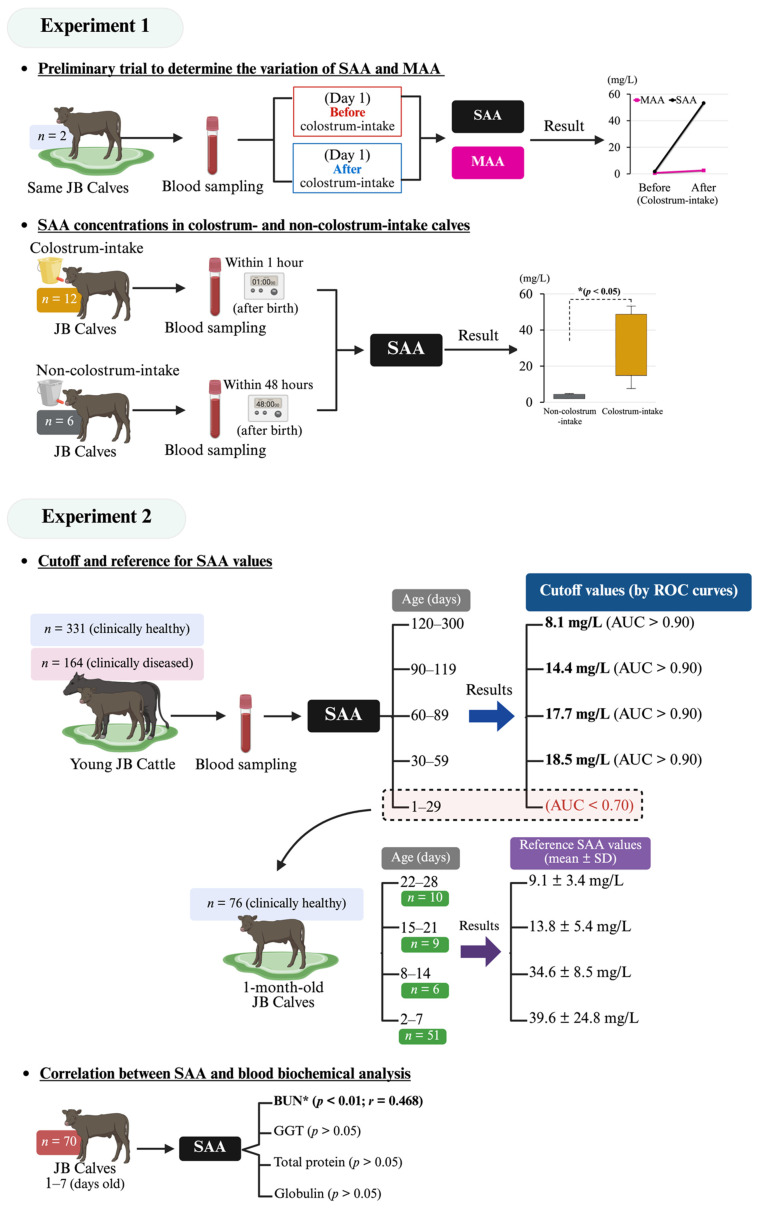
An illustration of the experimental design and summary of the results in this study. AUC, area under curve; BUN, blood urea nitrogen; *: *p* < 0.05; GGT, γ-glutamyltransferase; JB, Japanese Black; ROC, receiver operating characteristic; SAA, serum amyloid A.

**Table 1 animals-15-01239-t001:** Total number and distribution of clinically healthy and diseased young Japanese Black (JB) cattle used in the cut-off analysis by age group (days).

Health Status	Age (Days)					Total
	1–29	30–59	60–89	90–119	120–300	
Clinically healthy	94	32	63	62	80	331
Clinically diseased	39	18	20	23	64	164

**Table 2 animals-15-01239-t002:** Number of diseased calves in the present study based on the clinical diagnoses by the veterinarians.

Clinical Diagnosis (*n*)	Age (Days)					Total
	1–29	30–59	60–89	90–119	120–300	
Weak calf syndrome	9					9
Enteritis	22	2	5	8	14	51
Pneumonia	3	4	7	6	23	43
Bronchitis	5	12	7	5	15	44
Arthritis			1		1	2
Myositis				1	2	3
Bladder rupture				1		1
Cystitis				1	1	2
Urinary stone				1	7	8
Peritonitis					1	1
Total	39	18	20	23	64	164

## Data Availability

The original contributions presented in the study are included in the article. Further inquiries can be directed to the corresponding author.

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
