# Peer review of "Serum Amyloid A Concentrations in Young Japanese Black Cattle: Relationship with Colostrum Intake and Establishing Cut-Off Concentrations"

_animals, 2025, doi:10.3390/ani15091239_

Round 1
Reviewer 1 Report
Comments and Suggestions for Authors
You'll find my comments in the word-file added below!
In case of any ambiguities please send me a mail!

The English is not perfect and needs another view from a native English speaker - but the quality of the English language is not the main problem of the manuscript.
Author Response
Comments and Suggestions for Authors
P 1 L 20: I am sorry but the authors did not provide SAA cut-off contrations from birth to growth – for calves at an age below 30 days, respective values could not be calculated.
Thank you very much for this insightful remark. Accordingly, we have revised the sentence based on our obtained results as follows: “30 days after birth through the growing period” Page 1, Line 20.
P 1, L 37-39: No, sorry, but the results of this study do not allow to interprete unambiguously SAA concentations in JB calves – please rephrase this sentence.
Thank you for your comments. To present future possibilities based on the results obtained, we have modified the text as follows: “Evaluating SAA concentrations in JB calves, while considering factors such as age and colostrum intake, may contribute to improved calf productivity” Page 1, Lines 37-39.
P 2, L 51: It is suggested to replace “breeding cattle” by “beef cows”.
Thank you very much for this insightful remark. We have made the necessary changes accordingly. Page 2, Line 52.
P 2, L 58: From my point of view, especially the innate immune system of calves is well developed already after birth, but - and here I agree with the authors - not fully differentiated. If the authors agree, the sentence could be rephrased.
This part is simply a partial quote from previous publications on innate immunity in JB calves. We would like to keep it as is. Thank you for your understanding.
P 2, L 69-72: If I understood this sentence correctly, the approach of the authors is to measure SAA concentrations routinely in order to identify diseased animals? This appears to me questionable – the basis should be the daily visual assessment of the general condition, feed intake and clinical status and any lab tests should be performed to get additional information based on the clinical findings?!
We apologize for the lack of explanation. However, the approach is not to routinely measure SAA concentration, but rather to visually assess the daily general condition, feed intake, and clinical status. The goal is to calculate a cut-off value for SAA concentration to provide additional information on the inflammatory status based on clinical findings. Thank you for your understanding.
P 2, L 90: “Additionally, it has recently been reported that colostrum and calf serum SAA …”: please indicate the reference (is it [13]?) and specify the meaning of “colostrum” – do you mean amount of consumption, quality or something else?
Thank you for your remarks. Accordingly, we have added the reference to the text. We apologize for the confusion regarding “colostrum”, and have revised the sentences based on the referenced manuscript as follows: “Additionally, it has recently been reported that colostrum SAA and calf serum SAA are negatively associated”. Page 2, Lines 89-90.
P 3, L 125-126: It remains unclear how diseased animals were identified – based on results of blood biochemistry or based on clinical symptoms? Unclear: “diseased animals were requested to be examined for disease diagnosis” – what does that mean? Unfortunately, the results of disease diagnosis are not explained at all. In calves, infectious as well as non-infectious diseases are common. Having APP in the focus, this should make a difference. Were blood samples taken exclusively from acutely ill animals or also from calves suffering from disease since a longer period of time?
Based on my own clinical experience, most diseased calves at an age of 1-3 weeks have enteric problems, while in older calves bovine respiratory disease (BRD) is the major cause of treatments – is this also the case in JB calves? If so, does this shift of predominating diseases affects as well the SAA results of diseased calves in this study? May it explain the AUCs of ROC curves?
P 4, L 166: Thus, please provide information on the specific diagnoses of the 164 diseased calves used for Exp. 2!
We apologize for the lack of detailed information on normal and diseased calves. Accordingly, we have added and revised the sentences as follows: “The healthy young JB cattle included in this study were of various ages and showed no visible signs of clinical disease. They underwent routine metabolic profile tests (health checks) conducted at our clinic, with blood biochemistry analyses performed in the laboratory. In contrast, diseased animals were examined upon request by farmers for disease diagnosis. All animals were housed indoors, with feeding and management practices varying by herd. Suckling calves were fed either their mother's milk or a milk replacer, while post-weaning growing cattle were primarily fed roughage and a growing concentrate. Blood samples from healthy animals were collected from the jugular vein of each herd after the morning feeding. For diseased animals, blood samples were collected in the same manner from affected herds, following a veterinarian’s examination and clinical diagnosis. In this study, the blood sampling from all diseased calves was conducted between days 1 and 4 of illness.”. Page 3, Lines 123-136.
Additionally, we have provided information on the specific diagnoses of the 164 diseased calves in Table 2.
P 4, L 178: Here and in the whole manuscript: please have in mind that a calf can not be 0 month old neither can it live at 0 days of age (P 5 L 204). At calving, in the first minute of extrauterine life, the first day of life and the first month of life and the first year of life begins. Thus, please avoid any sentences or statements like “concentrations on day 0” …
We appreciate this insightful remark. Accordingly, we have replaced all instances of “day 0” with “day 1”.
P 9 L 265-268: “.. is the colostrum which significantly fluctuates in the first week of life and stabilizes after the third week …” – the meaning of this statement is unclear -what do the authors mean??
We apologize for this confusion. We have revised the sentence as follows: “Regarding blood SAA levels in newborn Holstein calves, one of the major factors affecting neonates in the first week of life is colostrum intake, which significantly fluctuates in the first week of life and stabilizes after the third week, although it is also influenced by environmental factors and age.” Page 10, Lines 281-284.
P 9/10: The discussion leaves me rather confused: are high serum SAA concentrations in the first weeks of age advantageous due to a potentially protective role ? If so, what does a cut-off value mean at all? There seem to be two aspects: the biological defense function on the one hand and the SAA concentration as an indicator for pathological inflammation/disease, on the other. If so, why did the author not measure other APP, such as haptoglobin/albumin etc. in the same samples which were analyzed for SAA?? Please, could the authors be so kind to sort this out!
We apologize for the confusion. The results of this study suggest that the cutoff value, which showed significantly high sensitivity between normal and diseased calves after one month of age, is an important indicator for the early detection and prognosis evaluation of inflammatory diseases in cattle production. Regarding the other APPs mentioned, we measured albumin concentration in this study, but found no correlation with SAA concentration. We plan to examine the relationship with other APPs, such as haptoglobin, in future research. Thank you very much for your understanding.

Reviewer 2 Report
Comments and Suggestions for Authors
General comments
The study refers to the concentrations of amyloid A in the serum of young Japanese black cattle, and the assessment of the concentration of SAA, whether it can contribute to calf productivity through colostrum intake status, and the determination of concentrations from birth to the growing period in this Japanese cattle breed.
The study is well designed with adequate methodology, presented results, statistical analysis, discussion and conclusion.
Special comments
lines 137-155: You stated that you measured the following parameters: blood sialic acid, glucose, free fatty acids, total cholesterol, total protein (TP), albumin, bilirubin, urea nitrogen, aspartate aminotransferase, γ-glutamyl transferase, calcium, magnesium, iron, and inorganic phosphorus, but only some were shown, as well as complete blood counts, serum vitamin A, and vitamin E.
Where are the other listed parameters (except for the association of SAA with TP, BUN, GGT and globulins) shown? Please show the results or delete sentences about CBC, Vitamins ( and E) and other biochemical parameters you didn't present.
Author Response
Comments and Suggestions for Authors
The study refers to the concentrations of amyloid A in the serum of young Japanese black cattle, and the assessment of the concentration of SAA, whether it can contribute to calf productivity through colostrum intake status, and the determination of concentrations from birth to the growing period in this Japanese cattle breed.
The study is well designed with adequate methodology, presented results, statistical analysis, discussion and conclusion.
lines 137-155: You stated that you measured the following parameters: blood sialic acid, glucose, free fatty acids, total cholesterol, total protein (TP), albumin, bilirubin, urea nitrogen, aspartate aminotransferase, γ-glutamyl transferase, calcium, magnesium, iron, and inorganic phosphorus, but only some were shown, as well as complete blood counts, serum vitamin A, and vitamin E.
Where are the other listed parameters (except for the association of SAA with TP, BUN, GGT and globulins) shown? Please show the results or delete sentences about CBC, Vitamins (and E) and other biochemical parameters you didn't present.
Thank you very much for this insightful remark. In this study, CBC and blood biochemistry tests were performed to evaluate their correlation with postnatal SAA concentrations. As a result, items for which no significant correlation was found were excluded. Therefore, a new statement explaining this has been added to the results section as follows: “No significant correlation was found between SAA and blood sialic acid, glucose, free fatty acids, total cholesterol, aspartate aminotransferase, calcium, magnesium, iron, and inorganic phosphorus, complete blood counts, or serum vitamin A, and vitamin E, and therefore, the results for these parameters are omitted.” Thank you for your understanding. Page 7, Lines 252-256.

Reviewer 3 Report
Comments and Suggestions for Authors
This study explores the variations in serum amyloid A (SAA) concentration in young Japanese Black (JB) cattle in relation to colostrum intake and establishes age-specific SAA cut-off values. Elevated SAA levels were found in calves that consumed colostrum compared to those that did not. The study also finds that SAA levels in calves are not directly absorbed from colostrum but are likely biosynthesized by the calves themselves as part of an innate immune response. Four age-specific cut-off concentrations were established for SAA, facilitating better health monitoring and management of JB calves from birth to 300 days old.
- How did variations in management practices across different farms affect the study results? Providing insights into the standardization of environmental and management conditions might help in interpreting the variability in SAA concentrations more accurately.
- The manuscript uses ROC curves to determine cut-off values. Given that AUC values vary with age, could you provide more insight into why the AUC for calves aged 0−29 days showed poor performance?
- Can you explore further how the apparent lack of correlation between SAA concentrations and globulin levels might impact the interpretation of colostrum intake efficacy or suggest alternate biomarkers that might reflect passive transfer more accurately?
- The study concludes that SAA concentrations in calves are not derived from colostrum but are biosynthesized. Could you elaborate on the mechanisms you propose for this biosynthesis? Are there specific triggers in the colostrum that initiate this synthesis, and how might this information affect the management of neonatal calves in terms of colostrum feeding strategies?
Author Response
Reviewer 3
Comments and Suggestions for Authors
How did variations in management practices across different farms affect the study results? Providing insights into the standardization of environmental and management conditions might help in interpreting the variability in SAA concentrations more accurately.
We appreciate this insightful remark. Unfortunately, based on the results of this field study, it is difficult to interpret the impact of differences in environmental and management conditions between farms on the variation in SAA concentrations. We would like to use your comments as a topic for future research. Thank you for your understanding.
The manuscript uses ROC curves to determine cut-off values. Given that AUC values vary with age, could you provide more insight into why the AUC for calves aged 0−29 days showed poor performance?
Thank you for your insightful comments. Unfortunately, we do not have a response to your comment at this time. As shown in Figure 3, large individual differences in SAA concentrations during the first week of life may be involved. Further detailed studies are needed. Thank you for your understanding.
Can you explore further how the apparent lack of correlation between SAA concentrations and globulin levels might impact the interpretation of colostrum intake efficacy or suggest alternate biomarkers that might reflect passive transfer more accurately?
We appreciate this insightful remark. While we currently have no suggestions regarding alternative biomarkers, we are continuing our research into the correlation between SAA concentrations and globulin concentrations when artificial colostrum is fed to postnatal calves. This investigation aims to help interpret the usefulness of colostrum intake. Thank you for your understanding.
The study concludes that SAA concentrations in calves are not derived from colostrum but are biosynthesized. Could you elaborate on the mechanisms you propose for this biosynthesis? Are there specific triggers in the colostrum that initiate this synthesis, and how might this information affect the management of neonatal calves in terms of colostrum feeding strategies?
Thank you for your insightful comments. The increase in SAA concentration in calves after ingesting colostrum has already been reported in calves other than JB. As stated in the text, it has been shown that pro-inflammatory cytokines in colostrum, such as interleukin (IL)-6, IL-1, and tumor necrosis factor a, stimulate the production of SAA in the liver of calves. The results of this study suggest that the same mechanism applies to JB calves. The interleukin (IL)-6, IL-1, and tumor necrosis factor a contained in colostrum may reflect the health and nutritional status of the mother cow. Therefore, improving the health and nutritional status of the cow before calving may reduce the incidence of diseases in the calf and contribute to successful calf management. Thank you for your understanding.

Round 2
Reviewer 1 Report
Comments and Suggestions for Authors
Please consider the attached file!

Author Response
Animals
Manuscript ID: animals-3466499
Serum Amyloid A Concentrations in Young Japanese Black Cattle: Relationship with Colostrum Intake and Establishing Cut-Off Concentrations
We thank the reviewer for the detailed and constructive comments that improved our manuscript. We revised the manuscript in accordance with the reviewers’ suggestions. We have marked the revised sections of the manuscript and the reviewers' responses in red. Professional English editing (Editage) has revised the paper to improve its quality.
P 1, L 26: it is suggested to replace “their influencing factors” by “affecting factors”.
Accordingly, we have revised “their influencing factors” to “affecting factors” Page 1, Line 26.
P 1, L 29: In the first review, I politely asked for avoiding to use the incorrect terms “day 0” and “0 month”. In the response letter, the authors stated that they replaced all instances. Even in the abstract, however, three times “day 0” or “0 month” is used (L 29, 32, 35). In the following manuscript, there are still so many incorrect that it makes me really wonder (e.g., Figure 1, Figure 5, L 184, L 218, L 278, L 324 etc.).
We sincerely appreciate this comment and sincerely apologize for this. We have re-checked and made the necessary changes accordingly.
P 1, L 30: here, GGT is described as gamma-glutamyltranspeptidase, later in the manuscript gamma-glutamyltransferase or gamma-glutamyltransferase. Please use gamma- glutamyltransferase (GGT) in the whole manuscript!
Thank you for this remark. According to the remark, we used “γ-glutamyltransferase (GGT)” in the whole manuscript consistently.
P 1, L 37-39: “Evaluating SAA concentrations in JB calves, while considering factors such as age and colostrum intake, may contribute to improved calf productivity.”
No, I am sorry, but the results of this study do not support this conclusion. It is suggested to omit this sentence. And this is already the key critical point in the whole manuscript from my point of view: unfortunately, it remains open whether high SAA concentrations in the first weeks of life represent a biological useful defense reaction or whether elevated levels are indicators of a pathological inflammation/disease — and it is unclear how a practitioner decides especially in very young calves whether a high SAA concentration is positive or negative in respect to the diagnosis.
We appreciate this feedback. We agreed to eliminate this sentence “Evaluating SAA concentrations in JB calves, while considering factors such as age and colostrum intake, may contribute to improved calf productivity.” Instead, we would like to insert the following new sentence as a conclusion of abstract to be considered based on the findings of this study “Clinical application of SAA concentrations in JB calves after one month of age may contribute to improving calf productivity”. Page 1, Lines 37-38.
P 2, L 54: it is suggested to replace “to prognosis diagnosis” by “to develop a diagnosis”.
We have made the following corrections in the English proofreading process “Furthermore, we demonstrated the clinical utility of our established cut-off value in farm practice, showing its applicability in diagnosing various diseases of JB cattle, such as postpartum metabolic disorders, bronchitis, and enteritis.” Page 2, Lines 51-54.
P 2, L 58: “... calves, especially JB calves, do not have well-developed innate and adaptive immune systems during the pre-weaning period [9,12].” No, I am sorry, but the innate immune system of calves is well developed already after birth, but - and here I agree with the authors - not fully differentiated. I would you like to suggest to rephrase the sentence in “... calves, especially JB calves, do not have fully differentiated innate and adaptive immune systems during the pre-weaning period [9,12].”
Thank you for this insightful remark. We agreed with this remark, and have rephrased the sentence accordingly. Page 2, Line 56.
P 2, L 64: it is suggested to replace “vaccines” by “vaccination protocols”.
Accordingly, we have replaced the part to address the ambiguity. Page 2, Line 62.
P 2, L 67: it is definitely not a general rule that calves lose weight during the period of weaning. Please put this statement into perspective.
Thank you for this comment. Weight loss has been revised to be expressed objectively as expected. Page 2, Lines 65-66.
P 2, L 80: please replace “influences” by “influence”.
Thank you for this correction. Accordingly, we have replaced the part. Page 2, Line 79.
P 2, L 91: “... when less protection from colostrum SAA is available [13], which may indicate a protective role for colostrum SAA shortly after birth.” Difficult to understand for me because authors argue that colostral SAA is not absorbed by the newborn calf?
Thank you for this comment. The description here will focus on the role of colostrum SAA, which we hypothesize as one of the possibilities based on suggestions from a previous report (13). Therefore, we would like to keep this sentence. Thank you for your kind understanding.
P 3, L 133/134: I assume that the authors meant “calf” and not “herd”?
Thank you for this remark. Accordingly, we have replaced the part. Page 3, Lines 128 and 130.
P 4, L 181: please replace “aged 2 28 days” by “aged 2 to 28 days”.
We have revised the section accordingly. Page 4, Line 176.
P 4, L 184: please replace “correlation ... was analyzed” by “correlations ... were analyzed”.
Accordingly, we have revised the sentence. Page 4, Lines 179-180.
P 8, Fig. 4: It is unclear, why same ranges were found for serum concentrations of total protein and globulins. From my understanding, globulins should be a proportion of TP having in mind that albumin is not mentioned here. Anything has to be wrong. From my experience, lgG ranges in the serum between 5 and 30 g/L — the figure in this paper suggests something totally different. Please explain!
Thank you for pointing this out. Generally, globulin concentration is the sum of alpha, beta, and gamma globulin, which is calculated by subtracting albumin concentration from TP. We have checked the globulin concentration as you pointed out, and confirmed that the concentration was accurate. Thank you for your understanding.
P 9: In Fig. 5, reference ranges are presented for serum SAA in calves within their first four weeks of life. First, did you check the values for normal distribution? Second, reference ranges are mostly presented as means + 2 SD to have the range, where 95% of all individuals fall in. Why did the authors choose a different approach?
The range of SAA concentrations shown in the Fig.5 is not a standard value but merely a reference value, showing the average value and standard deviation at each age, and even if it is not normally distributed, it is possible to grasp the average and its surrounding values, and we have included them because we believe they are meaningful as reference values in the clinical practice of JB calves. We would appreciate your understanding.
P 10 L 282: “... which significantly fluctuates in the first week of life and stabilizes after the third week ...”— the meaning of this statement is unclear -what do the authors mean? I asked this question already last time.
We appreciate your comment and apologize for the confusion. After reviewing the paper 13 cited here, we determined that the citation in the text was inappropriate, therefore we have deleted the pointed out section.
P 1 0, L 313-315: Which statements refer to MAA, which to SAA? Confusing!
Thank you for this comment. We apologize for any confusion. We have revised the description of MAA and SAA to read “MAA in colostrum” and “SAA in blood”, respectively. Page 10, Lines 306-309.
P 11, L 353-355: This is precisely the dilemma of the entire study, which leaves me back perplexed.
Thank you for this comment and we apologize for any confusion. To avoid any confusion, we have deleted this sentence from the text.
P 1 1, L 360-367:
“In conclusion, our results suggest that elevated SAA concentrations in 1-month-old JB calves are not absorbed from colostrum, similar to what has been observed in Holstein calves. Instead, they may be biosynthesized by the calves themselves as part of their biological defense mechanisms. The cut-off SAA concentrations for JB calves were 18.5 mg/L at 30—59 days of age, 17.7 mg/L at 60—89 days, 14.4 mg/L at 90—119 days, and 8.1 mg/L at 120—300 days. Calves that ingested colostrum had significantly higher SAA concentrations compared to those that did not. Additionally, SAA concentrations at 1—7 days of age did not correlate with GGT, TP, or globulin levels.”
This paragraph is completely incomprehensible to anyone who has not read the entire manuscript. The order of the statements has to be changed.
Thank you for this remark. In response, we have rearranged and improved the order of the sentences to make this conclusion easier to understand. Page 11, Lines 352-359.

Reviewer 3 Report
Comments and Suggestions for Authors
Agree to publish.
Author Response
Thank you very much for your decision.
Round 3
Reviewer 1 Report
Comments and Suggestions for Authors
Thank you so much for your efforts to revise the paper! I do not have further concerns. All the best and warm regards